# A Decade in Review: A Systematic Review of Universal Influenza Vaccines in Clinical Trials during the 2010 Decade

**DOI:** 10.3390/v12101186

**Published:** 2020-10-20

**Authors:** Brigette N. Corder, Brianna L. Bullard, Gregory A. Poland, Eric A. Weaver

**Affiliations:** 1School of Biological Sciences, Nebraska Center for Virology, University of Nebraska-Lincoln, 4240 Fair Street, Lincoln, NE 68503, USA; Brigette.corder@huskers.unl.edu (B.N.C.); bbullard@huskers.unl.edu (B.L.B.); 2Mayo Vaccine Research Group, General Internal Medicine, Mayo Clinic, Rochester, MN 55902, USA; Poland.Gregory@mayo.edu

**Keywords:** influenza, universal influenza vaccine, influenza vaccine, clinical trials, vaccine target, vaccine platform, adjuvant

## Abstract

On average, there are 3–5 million severe cases of influenza virus infections globally each year. Seasonal influenza vaccines provide limited protection against divergent influenza strains. Therefore, the development of a universal influenza vaccine is a top priority for the NIH. Here, we report a comprehensive summary of all universal influenza vaccines that were tested in clinical trials during the 2010–2019 decade. Of the 1597 studies found, 69 eligible clinical trials, which investigated 27 vaccines, were included in this review. Information from each trial was compiled for vaccine target, vaccine platform, adjuvant inclusion, clinical trial phase, and results. As we look forward, there are currently three vaccines in phase III clinical trials which could provide significant improvement over seasonal influenza vaccines. This systematic review of universal influenza vaccine clinical trials during the 2010–2019 decade provides an update on the progress towards an improved influenza vaccine.

## 1. Introduction

Globally, seasonal influenza virus epidemics are estimated to cause 3–5 million cases of severe infection and result in 290,000–650,000 deaths annually [1,2]. Mortality is increased in the elderly over 65 years, children under 5 years, and people in developing countries [3,4]. In the United States alone, influenza virus infects between 9.2–35.6 million people each year, leading to 140,000–710,000 hospitalizations [5]. These annual influenza epidemics result in an estimated total economic loss of $87.1 billion each year due to direct medical costs and indirect costs such as projected lost earnings and loss of life [6]. While the disease burden for seasonal influenza epidemics is substantial, this is significantly increased during influenza pandemics. For example, it is estimated that 24% of the worldwide population was infected during the 2009 H1N1 swine influenza pandemic [7].

A substantial challenge in the development of an effective influenza vaccine is the significant viral population diversity. The current influenza vaccine can be either trivalent or quadrivalent. The trivalent vaccine contains a H1N1, H3N2, and an influenza B strain, with the quadrivalent vaccine including both Yamagata and Victoria influenza B lineage strains [2,8]. The strains contained in the seasonal influenza vaccine are updated yearly to include those predicted to circulate in the upcoming influenza season. Although the current influenza vaccine is effective at reducing morbidity and mortality due to seasonal influenza infections [9], vaccine effectiveness estimates only range from 10 to 60% [8,10]. The vaccine effectiveness is lowest when there is poor antigenic match to the circulating influenza strains [8,11].

Developing a universal influenza vaccine (UIV) that improves cross-protection is a high priority. In 2018, the National Institute of Allergy and Infectious Disease (NIAID) released a strategic plan for the development of a UIV. This plan suggested that the vaccine should (1) be at least 75% effective against symptomatic influenza virus infection, (2) protect against group I and II influenza A viruses, (3) have durable protection that lasts at least 1 year, and (4) be suitable for all age groups [1]. Many strategies have been explored towards the creation of a UIV. This systematic review examines the universal influenza vaccine candidates that have entered human clinical trials in the last decade. For each clinical trial we examine the target protein of influenza, the vaccine platform used, and the inclusion of adjuvants. A summary of these trials will inform researchers on the progress towards a UIV.

## 2. Materials and Methods

We performed a systematic review of UIVs that have been tested in clinical trials over the past decade. We defined a UIV as a vaccine that aims to induce better cross-protection than traditional seasonal influenza vaccines. Seasonal influenza vaccines were defined as a vaccine which annually changes the wild-type viral strain component(s) depending on circulating influenza strains. We searched the ClinicalTrials.gov database for trials on influenza with a primary completion date after 1 January 2010 and a study start date prior to 31 December 2019. Search terms included “influenza”, “influenza A virus”, “influenza vaccines”, and “universal influenza vaccine”. Studies were excluded if the primary outcome was to assess interactions between influenza and other diseases or comorbidities. Drugs and interventions for improving influenza illness outcomes were excluded. Interventions for improving seasonal influenza vaccination rates were also excluded.

Eligible trials were analyzed in detail for information on clinical trial phase, target of vaccine, vaccine platform, adjuvant, if applicable, and results. If results were not posted on Clinicaltrials.gov, the PubMed and PMC databases were searched with any combination of the following: NCT number, name of vaccine, study ID number, investigators, or responsible party. Trials with unpublished data are reported. For further information about the vaccine, the responsible party’s website was searched with any combination of the following: NCT number, name of vaccine, study ID number, or investigators. All publications relating to the vaccine and clinical trial were considered. Conflicts over inclusion were resolved by all authors. Selected studies are reported in Figure 1.

Relevant information was extracted from the ClinicalTrials.gov database including phase, vaccine target, vaccine platform, and results. If information was not available, relevant publications were analyzed. A summary of this data is reported in Table 1. All data were analyzed using GraphPad Prism 8.2 software. Figures were designed in Adobe Illustrator 2020 (24.0.1).

## 3. Results

Seasonal influenza vaccines provide limited protection and are updated annually to incorporate circulating strains. A vaccine which induces broad cross-protection against influenza remains a top priority for the National Institute of Health (NIH). Here we report a comprehensive review of universal influenza vaccine (UIV) clinical trials that were active between January 2010 and December 2019. In the last decade, 69 clinical trials investigating 27 vaccines were performed (Figure 1). These trials include a variety of viral targets, vaccine platforms, and adjuvants to boost the immune response to vaccination. Table 1 reports a chronological summary for each UIV clinical trial. The unique ID for each trial is used for identification in subsequent figures. Importantly, several UIVs were tested in up to 9 clinical trials. Since the trends may be skewed by these vaccines, we differentiate between vaccines and total clinical trials throughout this paper.

### 3.1. Vaccine Targets

Influenza vaccines typically target specific viral antigens to maximize the immune response to vaccination. Vaccination aims to induce a strong adaptive immune response which results in both T and B cell activation. These immune cells produce cytotoxic T cells and antibodies which can protect against future infection. Vaccines targeting internal viral proteins, such as nucleoprotein (NP) and matrix 1 (M1), can induce strong T cell responses [122]. Viral surface (external) antigens, hemagglutinin (HA) and neuraminidase (NA), are targeted by neutralizing antibodies [123]. Traditionally, a robust antibody response has been the goal of influenza vaccination and has been the basis upon which vaccines have been tested and licensed [124,125]. However, these antibodies provide limited protection against divergent influenza strains. Since there are strengths for both internal and external strategies, many vaccines include multiple antigens to induce a strong humoral and cellular immune response. Over the past decade, both internal and external influenza proteins were utilized in UIV clinical trials (Figure 2). Other strategies which target whole virus or attenuated virus through gene deletion have also been investigated. However, recent vaccines have focused on external proteins, specifically HA.

#### 3.1.1. Internal Proteins

Internal influenza proteins are attractive vaccine candidates since they are more conserved than the external glycoproteins [124]. This may result in broader cross-protection induced by the vaccine. One example is FP-01.1, a peptide-based vaccine which includes several CD4+ and CD8+ T-cell epitopes conjugated to a fluorocarbon chain. These epitopes are derived from internal influenza proteins including NP, M1, polymerase basic 1 (PB1), and polymerase basic 2 (PB2). Four trials were performed in the past decade utilizing 328 participants 18–74 years old. One phase I trial demonstrated that vaccination with FP-01.1 induced strong cellular responses in 75% of participants with a median response of 243 spot forming cells (SFC)/million peripheral blood mononuclear cells (PBMC) as measured using IFNγ ELISpot assay [65]. This cellular response was activated against several heterologous H1N1 and H3N2 strains indicating broad cross-reactivity [65]. Another UIV targeting internal proteins is OVX836, which is a recombinant NP vaccine. In preclinical trials, vaccinated mice were protected against three lethal influenza A virus (IAV) challenges and induced stronger immunogenicity than wild-type NP alone [118]. Protection was further improved if mice were immunized with a combination of the seasonal inactivated vaccine and OVX836.

#### 3.1.2. External Proteins

In the past decade, 14 of the 27 vaccines in UIV clinical trials targeted external glycoproteins. Although the HA protein has a high amount of diversity in the globular head, the HA stalk region is more conserved [125]. Vaccines targeting the stalk region of HA have shown promise during development and are being investigated in several UIV clinical trials [109,110,112,115,116]. One such vaccine is cH8/1N1, H5/1N1, which utilizes a prime-boost immunization strategy to promote an immune response towards the HA stalk domain. The phase 1 trial included 65 individuals between 18 and 39 years of age. Two viruses were modified with chimeric HA containing a homologous HA stalk and heterologous HA heads. These were administered as live attenuated or inactivated vaccines and boosted with a heterologous HA head vaccine 85 days later [109]. An oil-in-water adjuvant, AS03, was included with the inactivated vaccine. After the prime vaccination, only the adjuvanted groups induced strong IgG antibody responses. However, all groups demonstrated 2.2 to 5.6-fold increases in HA stalk specific IgG antibodies after a heterologous boost. These H1 HA antibodies were cross reactive to H2, H9, and H18 HA, indicating broad cross-protection against group 1 HA [109].

Another vaccine utilized the full-length H5 HA protein in an oral recombinant adenovirus type 4 (Ad4) vectored vaccine, Ad4-H5-Vtn. Three clinical trials have enrolled 313 participants between 18 and 49 years of age to investigate this avian H5 influenza vaccine. Three immunizations with Ad4-H5-Vtn resulted in low seroconversion, 11% for vaccinees and 7% for placebo [46]. Participants were boosted with an inactivated H5N1 vaccine, which resulted in 100% seroconversion for vaccinees compared to 36% in the placebo group. Vaccination with Ad4-H5-Vtn induced a significant T cell response after a single vaccination with a median 232 SFC/million PBMC. No serious adverse events were reported although vaccinees experienced higher rates of self-limited abdominal pain (16.8% vs. 2.4%), diarrhea (19.2% vs. 4.9%), and nasal congestion (32.8% vs. 14.6%) compared to the placebo.

NA was included in a DNA vaccine, VGX-3400X [60]. The DNA vaccine included plasmids containing NA, HA, and M2e-NP from H5N1 avian influenza. The vaccine was administered intramuscularly to over 200 participants 18–55 years of age during 4 clinical trials [59,61,62,63]. No results have been posted to date. Interestingly, NA was only investigated in combination with other influenza proteins. Furthermore, besides the VGX-3400X vaccine, NA was only included in whole inactivated and VLP vaccine strategies. The NA protein should be further investigated for its cross-protective potential against influenza [126].

### 3.2. Vaccine Platforms

Antigens are presented to the immune system in different ways depending on the vaccine platform. Most seasonal influenza vaccines utilize attenuated or inactivated wild-type viruses. These viruses display the external influenza proteins and stimulate strong antibody responses [127]. Although this strategy has been utilized since 1945, it has consistently shown low efficacy for protection against mismatched influenza strains [10,128]. Therefore, a variety of vaccine platforms were investigated during the last decade to further improve influenza vaccination. Although many vaccine platforms have been investigated, no single platform has thus far been demonstrated to show superior protection against influenza.

#### 3.2.1. Viral Vectors

A common platform for UIV clinical trials is viral vectors (29.4%), which utilize viral machinery to package, deliver, or display the vaccine antigen (Figure 3). Viral vectors have been commonly used as molecular biology tools and are approved for several gene therapies [129]. One of these vaccines is MVA-NP+M1, which is a modified vaccinia virus Ankara (MVA) viral vector expressing the nucleoprotein (NP) and matrix protein 1 (M1) genes from an H3N2 influenza strain [15]. In the past decade, nine clinical trials investigated MVA-NP+M1 enrolling over 7200 participants 18 years or older. Results from these trials report an increase in T cell response to vaccination, which remained significant above baseline for 52 weeks in 50–59-year-old participants. However, the response was only significant for 12 weeks in subjects 60–69 years old and 3 weeks for participants over 70 years [15]. A subsequent trial using 6 healthy subjects reported no significant difference in T cell response 21 days post-vaccination [24]. The antibody response to vaccination was not reported. To further boost the immune response to MVA-NP+M1, a heterologous boost with a simian adenovirus viral vector ChAdOx1-NP+M1 was investigated [21]. In this study involving 72 participants, both the MVA-NP+M1 and ChAdOx1-NP+M1 vaccines were shown to boost T cell responses when administered individually or together. A heterologous boost, regardless of the order, increased T cell responses ~5-fold. Another study investigating this heterologous strategy reported a significant increase in T cell responses at day 14 after ChAdOx1-NP+M1 vaccination, but a decreased response by day 21 [83]. Vaccinees were boosted with MVA-NP+M1, which again increased the T cell response; however, this response was not significant 21 days following the boost vaccination.

A recent viral vectored vaccine is Nasovax, an intranasal adenoviral vectored vaccine. Though no results have been posted for this clinical trial, data presented at the 2019 World Vaccine Congress reported strong immunogenicity and protection [105]. Indeed, vaccination with Nasovax induced 100% seroconversion, which was maintained for over 1 year.

#### 3.2.2. Nanoparticles

The newest vaccine platform utilizes nanoparticles to deliver viral antigens [130]. One vaccine utilizing this method is VAL-506440 and VAL-339851, which are mRNA HA from H10N8 and H7N9 influenza strains delivered in a lipid nanoparticle (LNP) [92]. Two trials were performed utilizing 357 participants aged 18–64 years. For the H10N8 mRNA vaccine, vaccination resulted in mild to moderate systemic adverse events including injection site pain (76.7–93.1% vs. 5.7–11.1%) and myalgia (47.8–70.9% vs. 2.9–3.7%) compared to the placebo. Antibody responses were increased in a dose-dependent manner for the H10N8 LNP vaccine reaching 100% seroconversion at 100 µg compared to 5.7% for the placebo group. These levels remained seropositive (HAI ≥ 10) for 6 months after immunization. The H7N9 LNP vaccine induced strong antibody titers for all vaccine doses with 96.3% seroconversion for the 25 µg dose group. Participants vaccinated with the H7 vaccine displayed mild injection site pain compared to the placebo (43.3–80% vs. 5.6–13.9%).

Other nanoparticle vaccines include VRC-FLUNPF081-00-VP, which is a recombinant HA vaccine delivered in a ferritin nanoparticle, and VRC-FLUNPF099-00-VP, which is a HA stalk protein delivered in a ferritin nanoparticle. Although neither trial has posted results, influenza ferritin nanoparticle vaccines have shown strong immunogenicity in mice and ferrets during preclinical trials [112].

#### 3.2.3. Recombinant Protein

Another common vaccine platform utilizes recombinant protein of a viral antigen (26.5%). Due to low immunogenicity, recombinant protein vaccines typically require the use of an adjuvant to enhance the immune response to vaccination [131,132]. Panblok is a recombinant HA protein administered with a novel stable emulsion adjuvant. Four clinical trials were performed which enrolled 1264 participants 18–49 years old. In one adjuvant dose-dependent trial targeting H5 influenza, results demonstrated that all adjuvanted vaccines (3.8 µg, 7.5 µg, or 15 µg) increased seroconversion from 9% in the unadjuvanted group to 70% for participants who received an adjuvanted vaccine [76]. However, another trial targeting H7 influenza reported low seroconversion regardless of the adjuvant dose [78]. Despite low antibody detection using HAI assay, antibodies against H7 influenza were detected using ELISA. Passive transfer of these antibodies resulted in protection against lethal H7 challenge in mice. Additionally, these antibodies were cross-reactive to H1, H4, H14, H3, H10, and H15, indicating broad immunity against both group 1 and group 2 influenza [78].

#### 3.2.4. Peptide Based Vaccines

Some vaccines deliver conserved immunogenic peptides of viral antigens. One such vaccine is Flu-v, a peptide-based vaccine containing conserved epitopes from influenza A and B viruses and adjuvanted with Montanide ISA-51. Over the past decade, 4 clinical trials were performed involving 408 participants between 18 and 60 years of age. Vaccination with Flu-v increased IFN-γ cellular responses 2-fold but did not induce antibody responses as expected [52]. In another study, seronegative males were vaccinated with Flu-v and then challenged with H3N2 influenza virus [55]. Participants vaccinated with Flu-v showed reductions in viral load and symptoms as well as an 8-fold increase in IFN-γ cellular responses.

#### 3.2.5. Attenuated Virus

Nonstructural protein 1 (NS1) is an influenza protein that antagonizes the immune system by downregulating antiviral host proteins [133]. Several attenuated virus vaccines in clinical trials have deleted this viral gene to improve the immune response to vaccination. Both GHB11L1 and GHB16L2 are intranasal live attenuated viruses with the NS1 gene deleted, but neither clinical trial has published results from these studies. Matrix protein 2 (M2) is an essential structural viral protein for influenza replication. The M2SR vaccine includes a virus that lacks the M2 protein resulting in non-infectious viral progeny, essentially a single-cycle virus [98]. All three M2SR vaccine trials are currently active.

### 3.3. Adjuvants

An ideal UIV will provide highly effective and long-lasting protection. This can be difficult to achieve when targeting internal proteins or using poorly immunogenic vaccine platforms. Adjuvants are compounds that stimulate the immune system and improve vaccine efficacy [132]. This is commonly achieved by oil-in-water emulsions, which recruit immune cells to the site of vaccination [134]. Another common group of adjuvants are toll-like receptor (TLR) agonists. These adjuvants bind and activate cellular host pathways, which leads to increased immune activation [135]. New adjuvants continue to be discovered and explored, but few are licensed for use in the United States [136].

#### 3.3.1. Oil-in-Water Emulsions

Over the past decade, most adjuvants in UIVs have been oil-in-water emulsions (39%) (Figure 4). M-001 is a recombinant protein vaccine that contains common B and T cell epitopes from the HA, NP, and M1 influenza proteins. Seven trials were performed over the past decade, which enrolled 10,391 individuals over 18 years old. This vaccine was combined with an adjuvant, Montanide ISA 51VG, which increased IgG titers 50-fold against the M-001 protein [137]. Strong T cell responses to M-001 were shown for all groups regardless of adjuvant inclusion. A subsequent trial reported M-001 could be used as a stand-alone or priming vaccine for the seasonal influenza vaccine [31]. When compared to seasonal vaccination alone, participants primed with M-001 before seasonal vaccination showed elevated antibody responses for matched H1N1 (4-fold vs. 2.24-fold) and H3N2 (3.17-fold vs. 2.3-fold), but not influenza B (1.7-fold vs. 1.32-fold). Additionally, M-001 vaccination increased both CD4+ and CD8+ T cell responses to H1N1, H3N2, and influenza B strains compared to baseline. Another unpublished clinical trial reported that 70% of M-001-vaccinated participants had a 4-fold increase in HAI titers compared to 41% for the control group [31]. This vaccine has moved into phase III clinical trials and was scheduled for primary completion in May 2020.

Immunose Flu is an inactivated split vaccine with a novel lipid adjuvant, Endocine. The immunogenicity of Immunose Flu was not reported, but vaccination resulted in serious adverse events in 2 of 36 participants including erysipelas and gastroenteritis [99]. Mild to moderate adverse events were recorded in 88.9% and 85.7% of vaccinated participants compared to 55.6% in the saline placebo control group.

#### 3.3.2. TLR Agonists

Another common group of adjuvants are toll-like receptor (TLR) agonists. These adjuvants bind and activate cellular host pathways, which leads to increased immune activation [135]. An example is VAX125, which is a recombinant HA protein fused to the TLR5 ligand, flagellin. Four clinical trials were performed using 911 participants over the age of 18. Vaccine doses over 5 µg resulted in ~8-fold elevated HAI titers, 75% seroconversion, and 98% seroprotection rates for H1N1 influenza [40]. However, dose escalation over 8 µg and 12 µg was stopped due to serious adverse events [42]. Vaccine doses ≥1.25 µg resulted in an average 19-fold increase in HAI titer, 92% seroprotection, and 79% seroconversion against a matched H1N1 influenza strain [42].

Another TLR adjuvant is double-stranded RNA (dsRNA), which binds TLR3 and activates inflammatory pathways [138]. VXA-A1.1 utilizes this adjuvant by encoding dsRNA and an H1N1 HA transgene in a recombinant adenovirus type 5 (Ad5) vector. This oral vaccine has been studied in 4 clinical trials with 285 participants between 18 and 49 years of age. One trial reported increased antibody responses to matched H1N1 strains with an average of 7.7-fold increases in HAI titers and 29-fold increases in microneutralization titers after vaccination [84]. Vaccination resulted in mild side effects at similar rates to the placebo group. Phase 2 clinical trial participants were immunized with VXA-A1.1 or the seasonal QIV vaccine and then challenged with an H1N1 influenza strain [87]. Vaccination with VXA-A1.1 resulted in 48% protection compared to 38% with the seasonal vaccine.

#### 3.3.3. Alum

Interestingly, although alum is one of the most commonly used FDA-approved adjuvants, only one clinical trial in 2010 utilized this adjuvant [136]. HAI-05 is a recombinant H5 HA protein vaccine that is produced in a plant-expression system, *Nicotiana benthamiana* [72]. This trial enrolled 100 individuals between 18 and 49 years of age and investigated the dose response of HAI-05 with alum. Interestingly, any combination of HAI-05 (15, 45, and 90 µg) with alum resulted in minimal antibody titers while HAI-05 alone (90 µg) induced the greatest antibody response (6.4-fold increase). This suggests the HAI-05 induced low immunogenicity that was not improved by the addition of an adjuvant.

### 3.4. Clinical Trial Phases

In the US, new drugs and vaccines must complete four phases of clinical trials to be licensed and marketed for public use. Phase I trials investigate the safety and dosage of the vaccine. Typically, phase I trials have limited numbers of participants and do not assess efficacy due to low statistical power [139]. Phase II trials assess the dose response, efficacy, and side effects of the new vaccine. These trials include more study participants and can last longer than phase I trials. Occasionally, phases I and II can be combined into one clinical trial, phase I/II. Phase III trials include a large sample size and assess participants for vaccine efficacy and adverse reactions. At this point, the new vaccine or drug may be approved for the market [139]. Lastly, phase IV clinical trials involve post-marketing surveillance of the efficacy and safety of the new vaccine. Importantly, not all clinical trial results are reported or published. It is common for results to be posted several years after the completion of a trial (Figure 5). Over the past decade, only half of completed trials reported their findings (Figure 5E). This delay is consistent regardless of clinical trial phase (Figure 5D).

As expected, most UIV clinical trials performed over the past decade were phase I trials (57.4%) (Figure 5). Of the 27 vaccines, 11 have progressed past phase I (40.7%); however, only 3 vaccines (11%) have been tested in phase III clinical trials. The first phase III trial investigated Inflexal V, a trivalent adjuvanted virus-like particle (VLP) vaccine [70]. This study included 205 children between 6 and 36 months and was completed in November 2010 [69]. All participants were immunized with a single full dose (0.5 mL) or with two doses (0.25 mL) of the Inflexal V vaccine. Results suggest that both vaccine groups demonstrated improved seroprotection and seroconversion rates. Participants who received two 0.25 mL doses 4 weeks apart showed higher seroprotection rates for H1N1 (99.0), H3N2 (99.0), and influenza B (92.2). For H1N1 and H3N2, the two-dose regimen resulted in higher seroconversion and geometric mean titer (GMT) fold increases than the single-shot regimen. Half of participants from each group experienced non-serious adverse events including pyrexia, malaise, rhinitis, cough, otitis media acute, as well as adverse events at the injection site including erythema, induration, pain, or hemorrhage.

The second UIV tested in a phase III clinical trial was M-001. This vaccine is a synthetic recombinant protein containing common linear influenza epitopes [31]. As discussed above, the adjuvanted M-001 vaccine has shown promising immunogenicity and the phase III trial was scheduled for primary completion in May 2020 [31,137].

The third vaccine tested in a phase III clinical trial is NanoFlu. This vaccine is a recombinant HA protein delivered in a nanoparticle with a saponin-based Matrix-M adjuvant [107]. Although results for the phase II trial have not been posted, a press release from Novavax stated that NanoFlu induced superior HAI antibody responses against homologous and drifted strains compared to the seasonal influenza vaccine. A phase III clinical trial involving 2650 participants over 65 years of age was scheduled for primary completion in December 2019.

## 4. Discussion

This systematic review documents UIVs that were tested in clinical trials from January 2010 to December 2019. Although many papers have discussed strategies for UIVs, few review papers address the translation of UIV strategies to clinical trials [140,141]. This is the first systematic review of UIVs in clinical trials.

The definition of a “universal” influenza vaccine is highly debated [125,141]. In 2018, the NIAID announced that a UIV should (1) be at least 75% effective, (2) protect against group I and II IAV, (3) have durable protection that lasts at least 1 year, and (4) be suitable for all age groups [1]. Since this standard was put forward towards the end of the decade, our definition of a UIV remains broader than the NIAID requirements. Here, we have defined a UIV as a vaccine that aims to induce better cross-protection than seasonal influenza vaccines. Therefore, “supra-seasonal vaccines” which cover a large subset of influenza strains and vaccines against specific subtypes of influenza have been included in this analysis.

The influenza diversity targeted by each vaccine varied. Only 37% of universal vaccines were designed to protect against both influenza A and B viruses. Other strategies focused on IAV (22%) or a single subtype of IAV (41%). Importantly, no vaccines focused on influenza B virus (IBV) alone. Furthermore, the current NIAID requirements for a universal influenza vaccine do not require cross-protection against IBV. Notably, the CDC reports that IBV is responsible for 72% of influenza cases reported for children and young adults each year [142]. Overall, approximately 26% of annual influenza cases can be attributed to IBV [143]. The significant burden of IBV should be addressed in the design of universal influenza vaccines.

Some limitations to this review should be noted. First, information about clinical trials can be limited until the results are published. Specifically, not all clinical trial summaries include information on vaccine design and mechanism. In these cases, previous publications and press releases for the vaccines were consulted. Additionally, most results reported safety information and homologous vaccine efficacy, providing limited information on the cross-reactivity of each vaccine. Second, we searched clinical trials registered through ClinicalTrials.gov, which could potentially exclude some studies. There are other clinical trial databases such as EU Clinical Trials Register, however, the ClinicalTrials.gov database reports more accurate and updated information for clinical trials [144].

Despite limited information, this review provides a comprehensive summary of the UIVs tested in clinical trials. Indeed, this is the first comprehensive review to also discuss efficacy and trends in vaccine development for influenza. The field of influenza vaccine development is ever progressing. This is reflected in new vaccine targets and platforms such as HA stalk and nanoparticles. Researchers over the past decade have produced many promising influenza vaccines, each with strengths and limitations. The efficacy of a vaccine may induce strong protection against matched strains, but an effective UIV must induce strong cross-protection as well. This review identifies vaccines that report efficacy against matched strains alone. Importantly, these vaccines may provide cross-protection if delivered in combination with vaccines targeting other influenza subtypes. However, this would require further research and investigation.

## 5. Conclusions

Influenza virus remains a major global pathogen despite the general widespread use of seasonal vaccines due to varying efficacy to drifted strains. A UIV remains a top priority for the NIH and World Health Organization. This review provides an update on the progress towards a better influenza vaccine. With this information, researchers and clinicians can remain informed about the status and limitations of universal influenza vaccines.

## Figures and Tables

**Figure 1 viruses-12-01186-f001:**
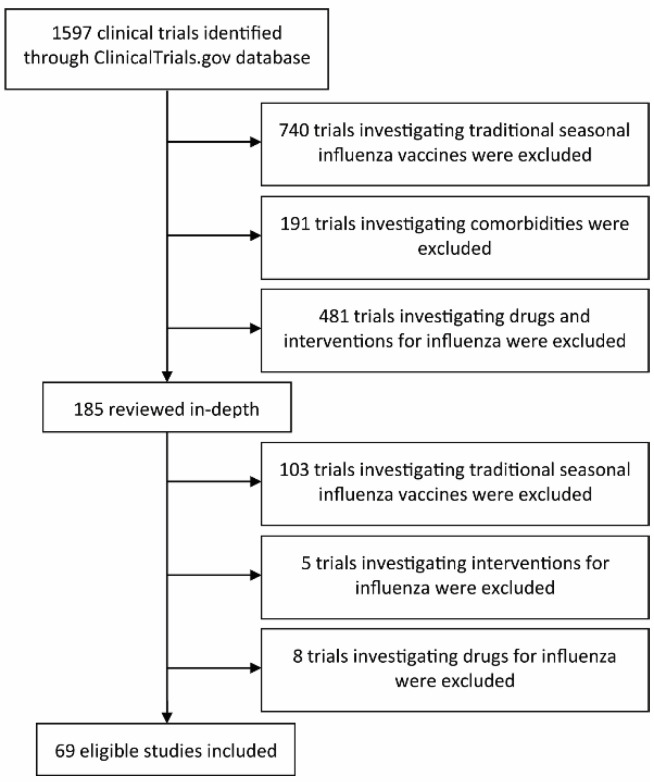
Study selection flow diagram. ClinicalTrials.gov database search resulted in 1597 clinical trials identified. After initial screening, 185 clinical trials were reviewed in depth. A total of 69 eligible universal influenza vaccine clinical trials were included in this review. See Materials and Methods for further information.

**Figure 2 viruses-12-01186-f002:**
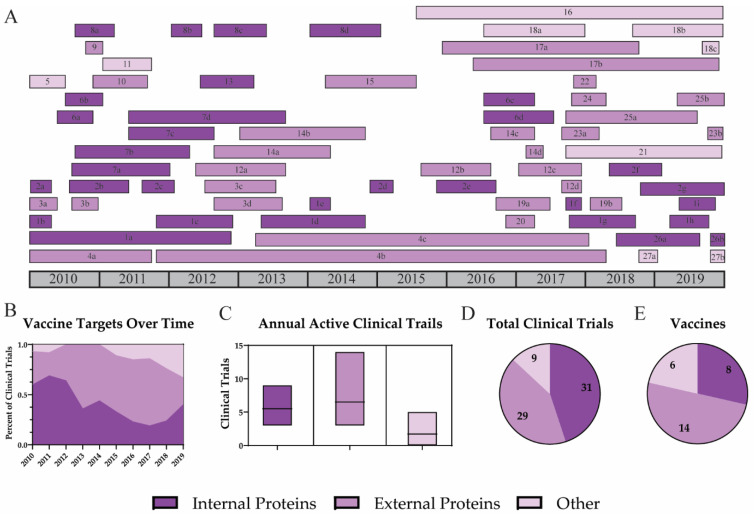
Targets for universal influenza vaccine strategies. A timeline of all universal influenza vaccine clinical trials is shown for vaccines which target internal proteins (dark purple), external proteins (purple), or other (light purple) antigens (**A**). Trends for vaccine targets are shown by the percent of active clinical trials each year (**B**). Average clinical trials for each target (**C**), the total number of clinical trials (**D**), and the number of vaccines directed against each target are reported (**E**).

**Figure 3 viruses-12-01186-f003:**
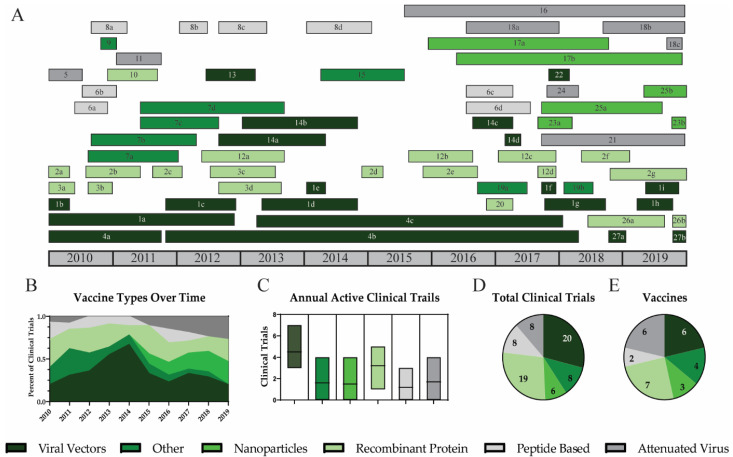
Vaccine platforms for universal influenza vaccines. A timeline of all universal influenza vaccine clinical trials is shown for vaccine platforms including viral vectors (dark green), other platforms (green), nanoparticles (lime green), recombinant proteins (light green), peptide-based platforms (light grey), and attenuated viruses (grey) (**A**). Trends for vaccine platforms are shown by the percent of active clinical trials each year. (**B**) The average number of active clinical trials each year (**C**), the total number of clinical trials (**D**), and the number of vaccines utilizing each platform during the decade (**E**) are reported.

**Figure 4 viruses-12-01186-f004:**
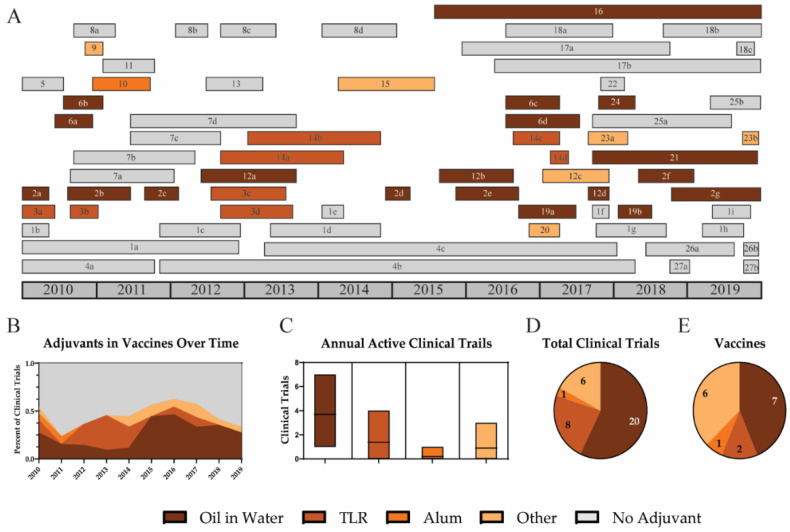
Adjuvants in universal influenza vaccine clinical trials. A timeline for adjuvants used for universal influenza vaccine clinical trials is shown for oil-in-water (dark brown), TLR agonists (brown), alum (orange), other adjuvants (light orange), and trials that did not use an adjuvant (grey) (**A**). Trends for the inclusion of various adjuvants are shown by the percent of active clinical trials each year (**B**). The average number of active clinical trials each year (**C**), the total number of clinical trials (**D**), and number of vaccines using each adjuvant during the decade are reported (**E**).

**Figure 5 viruses-12-01186-f005:**
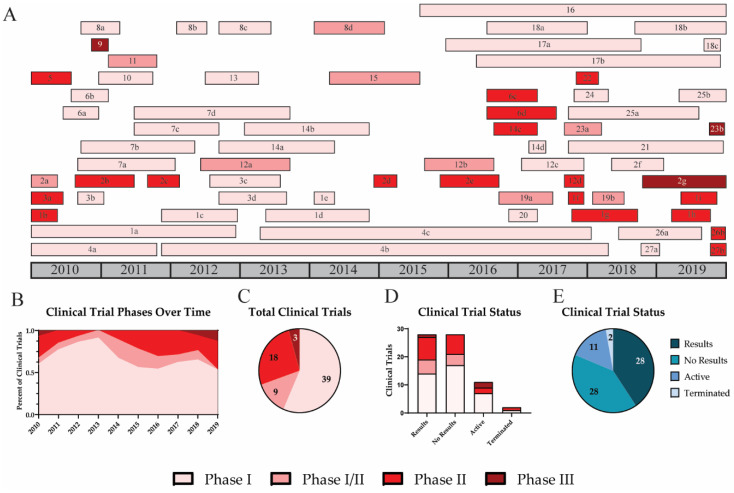
Clinical trial phases and results for universal influenza vaccines. A timeline for universal influenza vaccine clinical trials is shown for phase I (light pink), I/II (pink), II (red), and III (dark red) (**A**). Trends for vaccines in various clinical trial phases are shown by the percent of active clinical trials each year (**B**). The number of clinical trials in each phase is shown (**C**). Result status for trials in each clinical phase is shown (**D**). The total number of trials completed with results (dark blue), completed with no results (blue), active (grey blue), or terminated (light blue) is indicated (**E**).

**Table 1 viruses-12-01186-t001:** Characteristics of universal influenza vaccine clinical trials.

ID	Vaccine Name	Identifier	Phase	Target	Platform	Adjuvant	Adjuvant Type	Prime Boost	Results/Status	References
1a	MVA-NP+M1	NCT00942071	1	NP, M1	Viral Vector			Prime	Yes	[12,13,14,15]
1b	MVA-NP+M1	NCT00993083	2	NP, M1	Viral Vector			Prime	Yes	[16,17,18]
1c	MVA-NP+M1	NCT01465035	1	NP, M1	Viral Vector			Prime-Boost	No	[19]
1d	MVA-NP+M1	NCT01818362	1	NP, M1	Viral Vector			Prime-Boost	Yes	[20,21]
1e	MVA-NP+M1	NCT02014168	1	NP, M1	Viral Vector			Prime-Boost	Terminated	[22]
1f	MVA-NP+M1	NCT03277456	1	NP, M1	Viral Vector			Prime	Yes	[23,24]
1g	MVA-NP+M1	NCT03300362	2	NP, M1	Viral Vector			Prime-Boost	Terminated	[25,26]
1h	MVA-NP+M1	NCT03880474	2	NP, M1	Viral Vector			Prime	No	[27,28]
1i	MVA-NP+M1	NCT03883113	2	NP, M1	Viral Vector			Prime	No	[28,29]
2a	M-001	NCT01010737	1/2	HA, NP, M1	Recombinant Protein	Montanide ISA-51	Oil in Water	Prime-Boost	Yes	[30,31]
2b	M-001	NCT01146119	2	HA, NP, M1	Recombinant Protein	Montanide ISA-51	Oil in Water	Prime-Boost	No	[32,33]
2c	M-001	NCT01419925	2	HA, NP, M1	Recombinant Protein	Montanide ISA-51	Oil in Water	Prime-Boost	Yes	[31,34]
2d	M-001	NCT02293317	2	HA, NP, M1	Recombinant Protein	Montanide ISA-51	Oil in Water	Prime-Boost-Boost	No	[33,35]
2e	M-001	NCT02691130	2	HA, NP, M1	Recombinant Protein	Montanide ISA-51	Oil in Water	Prime-Boost-Boost	No	[33,36]
2f	M-001	NCT03058692	2	HA, NP, M1	Recombinant Protein	Montanide ISA-51	Oil in Water	Prime-Boost	No	[33,37]
2g	M-001	NCT03450915	3	HA, NP, M1	Recombinant Protein	Montanide ISA-51	Oil in Water	Prime-Boost	Active	[33,38]
3a	VAX125	NCT00966238	2	HA	Recombinant Protein	TLR5 + Flagellin	TLR Agonist	Prime	Yes	[39,40]
3b	VAX128	NCT01172054	1	HA	Recombinant Protein	TLR5 + Flagellin	TLR Agonist	Prime	Yes	[41,42]
3c	VAX161	NCT01560793	1	HA	Recombinant Protein	TLR5 + Flagellin	TLR Agonist	Prime-Boost	No	[43,44]
3d	VAX 161	NCT01658800	1	HA	Recombinant Protein	TLR5 + Flagellin	TLR Agonist	Prime-Boost	No	[45]
4a	Ad4-H5-Vtn	NCT01006798	1	HA	Viral Vector			Prime-Boost-Boost	Yes	[46,47]
4b	Ad4-H5-Vtn	NCT01443936	1	HA	Viral Vector			Prime	No	[48]
4c	Ad4-H5-Vtn	NCT01806909	1	HA	Viral Vector			Prime	No	[49]
5	GHB11L1	NCT01078701	2	ΔNS1	Attenuated Virus			Prime	No	[50]
6a	FLU-v	NCT01181336	1	NP, M1, M2	Peptide Based	Montanide ISA-51	Oil in Water	Prime	Yes	[51,52]
6b	FLU-v	NCT01226758	1	NP, M1, M2	Peptide Based	Montanide ISA-51	Oil in Water	Prime	Yes	[53,54,55]
6c	FLU-v	NCT03180801	2	NP, M1, M2	Peptide Based	Montanide ISA-51	Oil in Water	Prime-Boost	Yes	[56]
6d	FLU-v	NCT02962908	2	NP, M1, M2	Peptide Based	Montanide ISA-51	Oil in Water	Prime-Boost	Yes	[57,58]
7a	VGX-3400X	NCT01142362	1	HA, NA, NP	DNA			Prime-Boost	No	[59,60]
7b	VGX-3400X	NCT01184976	1	HA, NA, NP	DNA			Prime-Boost	No	[60,61]
7c	VGX-3400X	NCT01403155	1	HA, NA, NP	DNA			Prime-Boost	No	[60,62]
7d	VGX-3400X	NCT01405885	1	HA, NA, NP	DNA			Prime-Boost-Boost	No	[60,63]
8a	FP-01.1	NCT01265914	1	NP, M1, P1, P2	Peptide Based			Prime	Yes	[64,65]
8b	FP-01.1	NCT01677676	1	NP, M1, P1, P2	Peptide Based			Prime	No	[66]
8c	FP-01.1	NCT01701752	1	NP, M1, P1, P2	Peptide Based			Prime-Boost	No	[67]
8d	FP-01.1	NCT02071329	1/2	NP, M1, P1, P2	Peptide Based			Prime	No	[68]
9	Inflexal V	NCT01229397	3	HA	Virosome	Virosome	Virosome	Prime-Boost	Yes	[69,70]
10	HAI-05	NCT01250795	1	HA	Recombinant Protein	Alhydrogel	Alum	Prime-Boost	Yes	[71,72]
11	GHB16L2	NCT01369862	1/2	ΔNS1	Attenuated Virus			Prime	Yes	[73,74]
12a	PanBlok	NCT01612000	1/2	HA	Recombinant Protein	ASO3	Oil in Water	Prime-Boost	Yes	[75,76]
12b	PanBlok	NCT02464163	1/2	HA	Recombinant Protein	ASO3	Oil in Water	Prime-Boost	Yes	[77,78]
12c	PanBlok-H7	NCT03038776	1	HA	Recombinant Protein	AdVax	Δinsulin	Prime-Boost	No	[79,80]
12d	PanBlok-H7	NCT03283319	2	HA	Recombinant Protein	ASO3, MF59	Oil in Water	Prime-Boost	Yes	[81]
13	ChAdOx1-NP+M1	NCT01623518	1	NP, M1	Viral Vector			Prime-Boost-Boost	Yes	[82,83]
14a	VXA-A1.1	NCT01688297	1	HA	Viral Vector	dsRNA	TLR Agonist	Prime-Boost	Yes	[84,85]
14b	VXA-A1.1	NCT01761123	1	HA	Viral Vector	dsRNA	TLR Agonist	Prime	No	[86]
14c	VXA-A1.1	NCT02918006	2	HA	Viral Vector	dsRNA	TLR Agonist	Prime	Yes	[87,88]
14d	VXA-A1.1	NCT03121339	1	HA	Viral Vector	dsRNA	TLR Agonist	Prime	No	[89]
15	Avian Influenza VLP	NCT02078674	1/2	HA, NA	VLP	Matrix M1	Saponin Based	Prime-Boost	No	[90]
16	MER4101	NCT02500680	1	Whole Virus	Attenuated Virus	MAS-1	Oil in Water	Prime	Active	[91]
17a	VAL-506440	NCT03076385	1	HA	Lipid Nanoparticle			Prime	Yes	[92,93]
17b	VAL-339851	NCT03345043	1	HA	Lipid Nanoparticle			Prime	Yes	[92,94]
18a	M2SR	NCT02822105	1	ΔM2	Attenuated Virus			Prime	Active	[95]
18b	M2SR	NCT03553940	1	ΔM2	Attenuated Virus			Prime-Boost	Active	[96]
18c	M2SR	NCT03999554	1	ΔM2	Attenuated Virus			Prime-Boost	Active	[97,98]
19a	Immunose Flu	NCT02998996	1/2	HA, whole virus	Split Virion	Endocine	Oil in Water	Prime	Yes	[99]
19b	Immunose FLU	NCT03437304	1/2	HA, whole virus	Split Virion	Endocine	Oil in Water	Prime-Boost	No	[100]
20	NSV0001	NCT02955030	1	HA	Recombinant Protein	ND002	Other	Prime-Boost	No	[101]
21	D-SUIV	NCT03275389	1	Whole Virus	Attenuated Virus	ASO3	Oil in Water	Prime-Boost-Boost	Active	[102]
22	NasoVax	NCT03232567	2	HA	Viral Vector			Prime	Yes	[103,104,105]
23a	NanoFlu	NCT03293498	1/2	HA	Nanoparticle	Matrix M1	Saponin Based	Prime-Boost	No	[106,107]
23b	NanoFlu	NCT04120194	3	HA	Nanoparticle	Matrix M1	Saponin Based	Prime	Active	[108]
24	cH8/1N1, H5/1N1	NCT03300050	1	HA Stalk	Attenuated Virus	ASO3A	Oil in Water	Prime-Boost	Yes	[109,110]
25a	VRCFLUDNA081-00-VP	NCT03186781	1	HA	Ferritin Nanoparticle			Prime-Boost	Active	[111,112,113,114]
25b	VRCFLUNPF099-00-VP	NCT03814720	1	HA Stalk	Ferritin Nanoparticle			Prime-Boost	Active	[112,115,116]
26a	OVX836	NCT03594890	1	NP	Recombinant Protein			Prime-Boost	No	[117,118]
26b	OVX836	NCT04192500	2	NP	Recombinant Protein			Prime	Active	[118,119]
27a	GamFluVac	NCT03651544	1	Unknown	Viral Vector			Prime	No	[120]
27b	GamFluVac	NCT04034290	2	Unknown	Viral Vector			Prime	Active	[121]

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
