# Peer review of "A Decade in Review: A Systematic Review of Universal Influenza Vaccines in Clinical Trials during the 2010 Decade"

_viruses, 2020, doi:10.3390/v12101186_

Round 1

Reviewer 1 Report

Interesting and useful systematic review of universal influenza vaccines that have been submitted to clinical trials during the last 10 years. Vaccine prototypes are analyzed according their antigen target, vaccine platform, adjuvant and results. Data were extracted from the main clinical trial database. The summary provides an excellent overview of the evolution of vaccine technologies applied to achieve a cross-protective influenza vaccine. The figures and tables are very informative and easy to follow. The references are appropriate. The only aspect I miss is not providing information about the results of the clinical trials, at least of those whose information has been published. It would provide clues on the efficacy of the different antigens, platforms and adjuvants.

Author Response

Interesting and useful systematic review of universal influenza vaccines that have been submitted to clinical trials during the last 10 years. Vaccine prototypes are analyzed according their antigen target, vaccine platform, adjuvant and results. Data were extracted from the main clinical trial database. The summary provides an excellent overview of the evolution of vaccine technologies applied to achieve a cross-protective influenza vaccine. The figures and tables are very informative and easy to follow. The references are appropriate.

1. The only aspect I miss is not providing information about the results of the clinical trials, at least of those whose information has been published. It would provide clues on the efficacy of the different antigens, platforms and adjuvants.

The reviewer rightly points out that clear results would be beneficial for the interpretation of these clinical trials. We have complied the published information regarding initial immune correlates and reported this in the text of each section. Since most vaccines have not been tested in phase 3 trials, most of this information is limited to safety and immune correlates against matching strains or serotypes. This has been clarified in the discussion.

    a. Line 382: “Additionally, most results reported safety information and homologous vaccine efficacy, providing limited information on the cross-reactivity of each vaccine.”

2. To further clarify these results, we have added the following information regarding cross-reactivity and broad protection.

    a. Line 129: “This cellular response was activated against several heterologous H1N1 and H3N2 strains indicated broad cross-reactivity.”

    b. Line 147: “These H1 HA antibodies were cross reactive to H2, H9, and H18 HA indicating broad cross protection against group 1 HA.”

    c. Line 234: “Additionally, these antibodies were cross-reactive to H1, H4, H14, H3, H10, and H15 indicating broad immunity against both group 1 and group 2 influenza.”

Reviewer 2 Report

This manuscript provides systemic review of UIVs in clinical trials. I believe the authors spent lots of efforts in this review which discuss the efficacy and trends of UIVs. However, I have some suggestions.

  1. In Table I,is it possible to include the information if any trials using prime-boost strategy?
  2. In Figure 2, what are the “other” antigens used for UIV? The authors could summarize it.
  3. The authors only commented on seroconversion in the manuscript. Whether UIVs provide broad protection against other strains in any trial should be mentioned if possible.

Author Response

This manuscript provides systemic review of UIVs in clinical trials. I believe the authors spent lots of efforts in this review which discuss the efficacy and trends of UIVs. However, I have some suggestions.

1. In Table I, is it possible to include the information if any trials using prime-boost strategy?

We have added a Prime Boost column to Table 1. Strategies are reported as Prime, Prime-Boost, or Prime-Boost-Boost.

2. In Figure 2, what are the “other” antigens used for UIV? The authors could summarize it.

We agree that “other” should be clarified in the text. We have added a sentence starting on Line 112: “Other strategies which target whole virus or attenuated virus through gene deletion have also been investigated.”

3. The authors only commented on seroconversion in the manuscript. Whether UIVs provide broad protection against other strains in any trial should be mentioned if possible.

a. We agree that cross-protection is important when discussing UIVs. We have reviewed the published information for each trial and included information indicating cross-reactivity or broad immunity.

  1. Line 129: “This cellular response was activated against several heterologous H1N1 and H3N2 strains indicated broad cross-reactivity.”

  2. Line 147: “These H1 HA antibodies were cross reactive to H2, H9, and H18 HA indicating broad cross protection against group 1 HA.”

  3. Line 234: “Additionally, these antibodies were cross-reactive to H1, H4, H14, H3, H10, and H15 indicating broad immunity against both group 1 and group 2 influenza.”